# Accuracy and Precision Improvement of Temperature Measurement Using Statistical Analysis/Central Limit Theorem

**DOI:** 10.3390/s23063210

**Published:** 2023-03-17

**Authors:** Francisco Antônio Belo, Manoel Brasileiro Soares, Abel Cavalcante Lima Filho, Thyago Leite de Vasconcelos Lima, Marceu Oliveira Adissi

**Affiliations:** 1Research Group on Instrumentation and Control in the Study of Energy and Environment, Department of Electrical Engineering, Federal University of Paraiba, João Pessoa 58051-900, Paraiba, Brazil; 2Department of Mechanical Engineering, Federal University of Paraiba, João Pessoa 58051-900, Paraiba, Brazil; 3Department of Electrical Engineering, Federal Institute of Paraiba, Itabaian 58100-263, Paraiba, Brazil; 4Department of Electrical Engineering, Federal Institute of Paraiba, Cabedelo 58360-000, Paraiba, Brazil

**Keywords:** central limit theorem, accuracy improvement, temperature measurement

## Abstract

This paper describes a method for increasing the accuracy and precision of temperature measurements of a liquid based on the central limit theorem. A thermometer immersed in a liquid exhibits a response with determined accuracy and precision. This measurement is integrated with an instrumentation and control system that imposes the behavioral conditions of the central limit theorem (CLT). The oversampling method exhibited an increasing measurement resolution. Through periodic sampling of large groups, an increase in the accuracy and formula of the increase in precision is developed. A measurement group sequencing algorithm and experimental system were developed to obtain the results of this system. Hundreds of thousands of experimental results are obtained and seem to demonstrate the proposed idea’s validity.

## 1. Introduction

Very high accuracy was achieved by integrating technological advancement with measurement theory. It has major relevance to the economy, industry, agriculture, and health. In [1,2], a broad metrological approach is elaborated with the aim of universalizing concepts and methods but without imposing limitations.

If the random variations in the observations of an input quantity are correlated in time, the mean and experimental standard deviation of the mean traditional (µ,σ) may be inappropriate estimators [1]. In such cases, the observations should be analyzed by statistical methods specially designed to treat a series of correlated, random-varying measurements by the variances of Allan [3]. Compared to the conventional variance that assesses the variation around the mean value of the aggregate data surveyed, Allan Variance estimates variations by averaging measurements for different periods. This approach often leads to the possibility of directly distinguishing the different noise types (white, pink, random walk, blue, and violet) and to better convergence of the process of assessing their levels [4]. When there is only white noise, independent and uniformly distributed, it uses traditional variance, and even more, it is possible to apply the central limit theorem (CLT).

Precision is the closeness of agreement between the measured quantities obtained by repeated measurements of the same or similar objects under specified conditions. A high precision temperature based on [5] is one with precision greater than 0.001 °C. Measurement accuracy is the closeness of the agreement between a measured quantity value and the true value of a measurand [2]. The true value of the measurand is the value consistent with its definition. While precision is a measure of the uncertainty at a point on an axis with a random value, accuracy is related to the variation along the entire axis in relation to the true value.

In practice, there are a set of possible sources of uncertainty in a measurement, including [1]:(a)Incomplete definition of the measurand;(b)Imperfect realization of the definition of the measurand;(c)The sample measured may not represent the defined measurand;(d)Inadequate knowledge of the effects of environmental conditions on the measurement or imperfect measurement of environmental conditions;(e)Personal bias in reading analog instruments;(f)Finite instrument resolution or discrimination threshold;(g)Inexact values of measurement standards and reference materials;(h)Approximations and assumptions incorporated in the measurement method and procedure;(i)Variations in repeated observations of the measurand under apparently identical conditions.

The objective of measurement is then to establish the probability that this essentially unique value lies within an interval of measured quantity values based on the information available from the measurement quantity values, based on the information available from the measurement.

The concept of accuracy is discussed in other articles where the measurement is not temperature. According to the International Vocabulary of Metrology [2], the concept ‘measurement accuracy’ is not a quantity and is not given a numerical quantity value. A measurement is said to be more accurate when it offers a smaller measurement error. The geometric dimensions of small groups of measurements in the microlenses [6] and the method based on the microwave technique for thickness measurement of several thin coatings on a carbon composite [7] presented the accuracy of quantitative forms.

Precision is an observation at any point on the axis, where a set of observation data is selected, the mean is the assumed measurand value, and the precision is obtained from the variance calculated of this set [8]. One system is more accurate than another; its variation of the mean is smaller than the other system.

The performance of measurement is defined by its dynamics (measurement range, response time), accuracy (repeatability, precision, and sensitivity), and stability (tolerance for aging and harsh environments). Of these, accuracy is often considered to be the most important quality; it is also one of the most difficult to specify [9]. Specified accuracy usually includes repeatability and long-term stability. This allows for the calculation of the uncertainty of the calibration reference and the determination of the instrument’s overall accuracy. For a measurement group, the accuracy is (Δµ_th_)/(true value), similar to [6,7,9]. The mean of the difference between each measurement and the true value is Δµ_th_.

The theoretical aspects of the temperature concept are far beyond the specialized instrumentation literature. Temperature is an important parameter in every branch of physical science. The temperature measurement of a block surface depends on its properties, such as the thermal conductivity and free convection coefficient of the medium around the surface [10]. A simplified analytical model was used to predict the error in thermal properties and specific heat capacity [11].

An extremely high degree of temperature measurement accuracy can be achieved using a platinum thermometer. Electrical resistance measurements can be carried out with very high accuracy and with very little dissipated electrical power. The thermodynamic insulation conditions of a system with a thermometer immersed in a fluid are very high and have no external influence. In solving the heat transfer for this system, the only source of error is heat conduction, which is calculated to be very low and can be compensated. In an isolated thermodynamic system, a platinum sensor immersed in the system is at the same temperature as the system. The President of the Consultative Committee of Thermometry and Vice-president of the International Committee of Weights and Measures Division of Physics wrote the English version [12], authorized by the Thermometry Advisory Committee and approved by the CIPM. For the range 0 °C to 961.78 °C, in [12] is defined a reference function, polynomial of degree 9 of the relation of resistance with temperature within 0.13 mK.

The CLT is one of the most remarkable results in all of mathematics and accounts very largely for the importance of the normal distribution in theoretical investigations. The point of the theorem is that no matter the original distribution, the mean of a large enough sample will have a nearly normal distribution [13]. Many areas include CLT in its applications, such as computer science, psychology, political science, actuarial science [14], and engineering, economy, law, and probabilistic reasoning [15,16,17].

This paper presents a method that imposes the conditions of applying CLT in the measurement of temperature and determines the procedures and parameters needed to increase accuracy and precision, even in the presence of various systematic phenomena, resulting in very high accuracy. Oversampling theory is used to demonstrate the value of increasing the number of significant digits [18].

## 2. Materials and Methods

Consider the random variables *X*_1_, *X*_2_, … *X_k_* as the results of an experiment with *k* results and Independent and Identically Distributed (IID), with the mean and standard deviation *µ* and *σ* (parameters of the population), respectively. For each fixed *z*, the CLT is given as follows:(1a)$$\underset{k\to \infty }{\lim}P\left\{\frac{S_{k}-k\mu }{\sqrt{k}\;\sigma }\;\leq \;z\right\}=\emptyset \;\left(z\right)=\;\frac{1}{\sqrt{2\pi }}\int_{-\infty }^{z}e^{-u^{2}/2}du$$
(1b)$$\underset{k\to \infty }{\lim}P\left\{\frac{\bar{X_{k}}-\mu }{\;\sigma /\sqrt{k}}\;\leq z\right\}=\emptyset \;\left(z\right)=\;\frac{1}{\sqrt{2\pi }}\int_{-\infty }^{z}e^{-u^{2}/2}du$$
where *∅(z)* is the function of the normal distribution of the cumulative probability and *S_k_* = *X*_1_ + *X*_2_ + … + *X_k_*.

Consider that the random variables ${\bar{X}}_{N1}$, ${\bar{X}}_{N2}$, … ${\bar{X}}_{Nm}$ are the results of an experiment with m results and IID, and each result is an average of *N* repeated experiments with the mean and standard deviation ${µ}_{{\bar{X}}_{N}}$ and $\sigma_{{\bar{X}}_{N}}$ (population parameters), respectively. The mean property was ${µ}_{{\bar{X}}_{N}}=\mu$. For each fixed *z*, CLT is given as follows:(2)$$\underset{m\to \infty }{\lim}P\left\{\frac{{\left({\bar{S}}_{N}\right)}_{m}-m\mu }{\sqrt{m}\sigma_{{\bar{X}}_{N}}}\;\;\leq \;z\right\}=\emptyset \;\left(z\right)=\;\frac{1}{\sqrt{2\pi }}\int_{-\infty }^{z}e^{-u^{2}/2}du$$
where *∅ (z)* is the function of the normal distribution of the cumulative probability and ${\left({\bar{S}}_{N}\right)}_{m}$ = ${\bar{X}}_{N1}$+${\bar{X}}_{N2}$,…+ ${\bar{X}}_{Nm}$.

To apply the CLT to the mean of samples, consider each random variable as a mean of N elements of an *m* × *N* repeated experiment (${\left(\bar{X_{N}}\right)}_{1},{\left(\bar{X_{N}}\right)}_{2}\ldots \ldots {\left(\bar{X_{N}}\right)}_{M}$). The *S_K_* above is given:(3)$$S_{k}=X_{1}+X_{2}+\ldots +X_{k}=(X_{11}+X_{21}+\ldots +XN_{1})+(X_{12}+X_{22}+\ldots +XN_{2})+\ldots +(X1_{M}+X2_{M}+\ldots +X_{\mathrm{NM}})=\;N\times \left({\left(\bar{X_{N}}\right)}_{1}+{\left(\bar{X_{N}}\right)}_{2}\ldots +{\left(\bar{X_{N}}\right)}_{m}\right)=N\times N\times {\left({\bar{S}}_{N}\right)}_{M}\;e\;K=N\times m$$

Substituting this transformation into Equations (1)–(3) yields
(4)$$\underset{M\to \infty }{\lim}P\left\{\frac{N\times {\left({\bar{S}}_{N}\right)}_{m}-m\times N\mu }{\sqrt{m\times N}\sigma }\leq z\right\}=\emptyset \left(z\right)=\frac{1}{\sqrt{2\pi }}\int_{-\infty }^{z}e^{-u^{2}/2}du$$
(5)$$=\frac{N}{\sqrt{N}}\underset{m\to \infty }{\lim}P\left\{\frac{{\left({\bar{S}}_{N}\right)}_{M\;}-M\mu }{\sqrt{m}\sigma_{{\bar{X}}_{N}}}\;\;\leq \;z\right\}=\emptyset \;\left(z\right)=\;\frac{1}{\sqrt{2\pi }}\int_{-\infty }^{z}e^{-u^{2}/2}du$$
(6)$$\sigma_{{\bar{X}}_{N}}=\sigma_{N}=\;\frac{\sigma }{\sqrt{N}}$$
where *σ_N_* is the standard deviation of the population of all N elements mean of N repeated experiments with the population parameters ${µ}_{{\bar{X}}_{N}}$ and $\sigma_{{\bar{X}}_{N}}$.

If $\hat{\emptyset }$ is an estimator of parameter θ (mean *μ* or variance *σ*) of the random variables X_1_, X_2_, … X_N_, the sample mean $\bar{X}$ and sample variance *S* are unbiased estimators of *μ* and *σ*, respectively. With a biased estimator, according to [19], the mean squared deviation of the estimator MSD ($\hat{\emptyset }$) is
(7)$$MSD\left(\hat{\emptyset }\right)=E{\left(\hat{\emptyset }-\theta \right)}^{2}=V\left(\hat{\emptyset }\right)+bias^{2}$$

The bias is the mean variation along the axis, that is, *Δμ*. The mean squared deviation (MSDS) is its precision for any sample. The CLT variance would be a random variable (MSDS, i.e., *S*) but a population parameter (MSDP, i.e., *σ*). The condition for the application of CLT is that the bias is much smaller than the variance.
*Δμ* << *σ*(8)

The precision value does not depend on the number of samples when the variety of repetitive measurements is within the resolution of the measuring instrument. When calculating the averages, the number of significant digits increased. Applying CLT increases the precision and significant digit number of the mean, thereby increasing the precision and accuracy.

The precision is obtained from a set of measurements. The measured value is represented by the mean, and the precision is obtained by the variance of the set of measurement data. The temperature measurement is obtained from a set of measurements representing its thermodynamic state. An isolated system with no change in its thermodynamic state is at the same temperature.

Any set of measured data from an isolated system can be used to determine its temperature. The only source of error would be the small electrical voltage that would exist across the resistive sensor to produce the Joule effect. This electrical power can be made very small by the very small electrical voltage across it and still be compensated for by temperature control.

The CLT is applicable when there is no bias and when the measures are IID. These conditions will be validated experimentally below. Even if a distribution of the random variable is not normal, the distribution of the mean is normal. The practical importance of the CLT exists in that for a large but finite value of K, the cumulative distribution function (CDF) of the sum of SN is close to Ø(*z*) in the equation. The meanings of “large” and “close” are not easy to specify as a value of K nor as a degree of proximity. This depends on the nature of the common distribution [8]. This will depend on the instrumentation system and will be validated experimentally.

Oversampling frequency theory explains the increase in the number of significant digits when using the sampling mean. Consider each measurement made by the Analog Digital Converter (ADC) with a sampling frequency of *f_os_*. Sequential groups of *N* measurements and the mean values of these groups were calculated. These sequential groups of *N* consecutive measurements correspond to the sampling frequency *f_s_*. Thus, *f_os_* is the oversampling frequency of sequential means at *f_s_* rate. From these considerations and from [18], the variation in the significant digits is obtained as
*f_os_* = 4*^w^* × *f_s_*(9)
where *w* is the number of additional ADC resolution bits, *f_s_* is the original sampling frequency, and *f_os_* is the oversampling frequency.

### Experimental Methods

An instrumentation and control system maintains the temperature of the sample to be uniform, constant, and isolated from the external environment. This system drastically reduced the eight uncertainty sources related to the measurand. The block diagram in Figure 1 illustrates the operation of the system.

Internal metal blocks were embedded inside a sealed cavity into electronic transduction circuits and temperature measurement sensors. The cavity was fully filled with mineral oil, which was used for the cooling and electrical isolation of the power transformers. Temperature measurements of the sample’s environment were carried out using a temperature measurement system, and three measurements were used as a process variable to control the environment temperature evenly. Before and after this region, temperature measurements of the process variables also exist. All of these were based on a PT-1000 sensor (Platinum electrical resistance sensor, where the resistance is 1000 ohm when the temperature is zero degrees Celsius). The metal block was then heated or cooled using Peltier cells. Through ADC, the temperature data of the electronic transducers are read. Through a proportional-integral-derivative (PID) control algorithm, it sends the appropriate output signals to a Digital Analog Converter (DAC). Power Drive couplers are linear electronic circuits that use metal-oxide-semiconductor field-effect transistors (MOSFETs) to achieve DAC signals with powers adequate for Peltier cell operation. The PID controller operates through ADC, DAC, Power Drive, and Peltier cells to maintain the fluid environment in the cavity at a given temperature with high accuracy and precision. System diagram in Figure 1 shows the control algorithm and its parameters Td, Ti, and Kc (derivative, integral, and proportional gain terms, respectively). The output is buffered by a Power Drive and then goes to the Peltier modules.

In the temperature-controlled environment, there are three sensors that measure the temperature of the oil environment and are used to control the entire sample environment with high accuracy and uniformity. A fourth sensor is used for measurement only. The four thermometers are calibrated in another environment. The results being equal in this environment confirm that the measurement value is equal to the control set point value. When using the measurement sensor in an immersion measurement in an isolated thermodynamic system with uniform temperature, the temperature of this system will be measured with high accuracy. When the measurement sensor uses the measurement in this environment, it would be used as a calibration.

Figure 2a shows the metallic block with the Peltier cells outside. Figure 2b shows the cavity environment in which insulating oil, electronic transduction, sensors, and connection wires are placed. Helical and straight steel tubes (316 L) were cast in aluminum and placed inside the blocks. Each metallic block had external thermal insulation, with only the heat transfer path through the Peltier cells. The electronic power is filtered in such a way as to avoid bias (systematic effect) due to electrical effects on electronic transduction and control.

The helical shape of the steel tube was used to reduce the input and output effects on the ambient sample. Da Silva presented a theoretical and experimental study of this system [20]. To ensure a uniform temperature throughout the cavity, multiple input and multiple output PID controls were also used for this system.

The use of oil in the cavities (in which the electronic transducers, sensors, and connection wires are placed) protects the electronic elements from a breakage at the connections because of the condensation of water in ambient air in the absence of oil (breakage due to oxidation of the parts). It limits noise owing to these phenomena and allows for the long-term operation of the system and uniformity of the temperature in the environment. The sealed cavity, in an environment of high reproducibility and high temperature, contributes to the reduction in the systematic phenomena of electronic transduction due to ambient temperature, humidity, and pollution. The distinctive features of the electronic circuit with thermal drift compensation also contribute to its high accuracy and precision. The patent pending this system is presented in [21].

Figure 3 shows the electronic temperature measurement circuit. This circuit is based on the patent granted in 2020 [22], which had 10 years of analysis by the institution that conceded this right. As a patent granted, its reading is in the public domain, and its characteristics can be verified. Its behavior was experimentally demonstrated under rigorous conditions and applications [23]. The high gain of the circuit is attributed to the first differential amplifier, and the *C_D_* capacitor contributes to eliminating thermal drift. Computer simulation (TL084) showed that the output signal variation due only to ambient temperature variation, relative to 1 degree Celsius of the bridge’s sensor (PT 1000), was around 5.1 × 10^−6^ °C. The experimental results showed that the controlled environment is at a uniform temperature and with high accuracy, with a variation smaller than 0.001 °C. The resultant variation shall be much better than this value. The environment in which the transduction and sensors are located is an environment controlled by these same thermometers, which further contributes to high accuracy.

The description of the accuracy and precision of the platinum sensor and electronic apparatus are given in the form integrated. Through this, the error and uncertainty analysis are estimated. ITS-90 is defined in terms of the temperature dependence of the electrical resistance of standard platinum resistance thermometers (SPRTs). The practical realization of SPRT has been described [24,25]. Furthermore, they include an explanation of the scale definition, describe the different types of SPRTs used, give advice on the use and calibration of SPRTs, and conclude with a brief summary of the sources of uncertainty. They coincide very much with the global electronic apparatus in this paper. ITS-90 specifies a set of fixed points (melting, freezing, triple, or boiling points of various pure substances, which are used to calibrate SPRTs in 11 temperature sub-ranges within the overall range from 13.8033 K to 1234.93 K. These ranges can be used to calibrate the complete system.

Resistance measurements in platinum thermometry are usually made using automatic low-frequency resistance bridges. The major sources of uncertainty associated with the resistance measurements include the reference resistor value and its stability and the self-heating of the SPRT due to the sensing power. Sinusoidal and square-wave are two basic types of thermometry low-frequency resistance bridges in common use. The bridges typically have a resolution of six to nine digits and a specified uncertainty in resistance ratio between 2 × 10^−8^ and 4 × 10^−6^, corresponding to equivalent uncertainties in temperature measurements ranging from about 5 µK to 1 mK. The resolution of a bridge is determined by a combination of the thermal noise generated in all resistances and amplifiers; a quantization error due to the conversion of the signal to digital reading. Noise in the resistance measurement may also arise from electromagnetic interference (EMI). Other noises are mechanical vibrations as transformer cores lose magnetic permeability, a source of uncertainty in resistance measurements for high resolution or differential measurements (e.g., fixed-point comparisons and self-heating measurements). The magnitude of the resulting uncertainty can be inferred from the standard deviation of residual error in the calibration curves for bridges.

Measurements with resistance thermometers necessarily involve passing a current through the thermometer-sensing element, “self-heating”. As a result, the sensing element warms to a slightly higher temperature than that of the object or environment whose temperature is to be measured. The measurement current must be chosen to achieve good resolution without excessive self-heating. A dissipation of 25 μW at 0 °C is used, and the self-heating effect is typically 0.0002 °C. With modern resistance bridges, a temperature resolution of <0.0001 °C is easily achieved. Self-heating usually has a typical variation with temperature.

Experimental sources of uncertainty are due to impurities. Impurities cause irreversible changes in the resistance-temperature dependence of SPRTs and are a main cause of long-term drift. Static temperature-measurement errors are caused by the sensing element of the SPRT not being in direct thermal contact with the object of interest. The temperature indication of the thermometer is, in practice, affected by the many thermal conductances between the sensor and neighboring objects. A low radial thermal conductance also means a high self-heating effect. The immersion characteristic can be used experimentally to estimate the residual errors. A thermometer is sufficiently immersed when there is no detectable change in the indicated temperature with additional immersion in a constant-temperature environment.

Most of the topics described were performed. Manganin resistance was used to improve the reference resistor value and its stability. A bridge up to 400 Hz square wave was used, with a slew rate of amplifier much bigger than necessary. The power source was based on Voltage Reference sources (REF 02 and REF 10) with a drift minor that 8 ppm/°C. Power supplies (bridge and electronic apparatus) were isolated from the network electric to obtain high accuracy. Self-heating minor that 10 μW was used. The conduction and immersion effects are minimized due to the thermodynamic system of the sensor. The electronic control of temperature puts the sensor at a temperature very near the system. The power delivered to the sensor is withdrawn from the system by the temperature controller.

The electronic transduction circuits compensate very well for drifts due to the external environment (temperature, humidity, electric network). It could improve the random noise with the improvement of the filter of the power source, with an active filter, ADC, and DAC of bigger resolution and velocity.

The random noise obtained experimentally was approximately 0.001 °C, with values obtained from various sampling averages at different times. The value of the measurand would be 20.000 °C, with a precision of approximately 0.001 °C. The mean variations during all 11 days of measurements were much smaller than 0.001 °C, but this value limits the accuracy of the measurement to 0.001 °C. Greater accuracy will be possible with greater precision obtained by CLT.

An experimental procedure that was carried out to know about the accuracy was to use an oven and obtain the response (bridge plus electronic apparatus) with the temperature variation. A manganin reference resistor replaced the sensor. For a change of 50.0 °C, the response change was 0.1 °C which corresponds to 0.002 °C/°C. Considering only the effect of REF 02, without error due to switching, the expected variation would be around 0.0004 °C. This test was done before improving the power supply based on REF 02 and REF 10. Note that with the thermal control, which was 0.001 °C. This value would be 2.0 × 10^−6^ °C.

The programmable automation unit is based on the Compact Field Point controller manufactured by NI (National Instruments), with an intelligent communication interface (cFP-2000) and input and output modules (cFP-AI-110 (16-bit ADC) and cFP-AO-210 (12-bit DAC)).

## 3. Results

After calibration, the mean value is the measurand’s assumed value, and the mean squared deviation is its precision for any sample. The analysis was based on a large number of measurements (hundreds of thousands of data points) that were performed uninterruptedly for 11 days, with a measurement interval of 4 s. Different individual groups of 10, 50, 100, 200, 500, 1000, 5000, and 10,000 sample sizes were used, and in any of these groups, the precision was 0.001 °C, and the mean was 20.000 °C. These groups were randomly selected several times. Other measurements were performed several times within a time interval of up to several months. Experimental results from hundreds of thousands of measurement data points seem to validate the proposed method in the presence of the most varied effects due to systematic phenomena applied to the measurement of the temperature.

Random variables are identically distributed if they have the same probability distribution. These acquisitions are uniformly distributed as they had the same precision (the various datasets taken with the value of *N* of 10, 20, 50, 10, 500, 1000, and 10,000 showed the same precision). They are independent if a measurement on the same channel is independent of another. Two data acquisitions made by the analog-to-digital converter are independent because one acquisition is made after another, and their result is unrelated to the previous measurement. The other condition, the CLT, is applicable when there is no bias and it is due to systematic compensation.

The diagram in Figure 4 explains the methodology of the application of CLT (an explicative example was taken for *N* = 20 of a universe of more than 100,000 measurements). Several sets of *N* elements formed a class of *M* sets. This class is transformed into a set where the variance and mean are obtained from each set of *N* elements, creating a set of *M* means and *M* variances. These are the means and variances of the classes of *M* sets, with each set containing *N* elements. The CLT states that the sample mean tends toward the population. Resistance measurements in platinum average, with the width of the variance given by *σ/N*, as *M* increases.

The experimental procedure is applied according to the following algorithm:(a)Several sets of samples are chosen, with each set having *N* measurement data points (the values of *N* are 10, 20, 50, 100, 200, 500, and 1000);(b)The average of each of the *N*-samples groups above is taken, and several sets of *M* elements are selected, where each element of the sets is the average of these *N* data (the values of *M* are 10, 20, 30, 40, 50, 60, 70, 80, 90, and 100);(c)While increasing *M*, asymptotic behaviors are studied to verify the CLT;(d)Several sets of samples are chosen, with each set having *N*_1_ data (the values of *N*_1_ are 10, 20, 50, 100, 200, 500, and 1000), and each data point is the average of *N* according to item 1;(e)The average of each of the *N_1_* samples described above is obtained. Several sets of *M*_1_ elements are selected, where each element of the sets is the average of the *N*_1_ data (the values of *M*_1_ are 10, 20, 30, 40, 50, 60, 70, 80, 90, and 100);(f)While increasing M, the asymptotic behaviors are studied to verify the CLT;(g)Repeat 4 using *N*_2_ = *N*_1_ and *M*_2_ = *M*_1_.

Let *N* (number of measurements of an average) and *M* (number of averages) be the values chosen to apply the CLT. The flowchart in Figure 5 describes obtaining a value of greater accuracy and precision.

Figure 6 shows the results for *N* = 10, *N* = 100, and *N* = 500 with the variation in *M*. The curves seem to show the beginning of asymptotic behavior when *M* = 30. Table 1. shows the asymptotic behavior of the variations in the experimental means relative to the ideal values, with the growth in *M* for different values of *N*. Figure 7 shows the variation of the mean beginning at *M* = 40.

To visualize the asymptotic behavior of the variance and mean, Figure 8a,b shows the behavior with *N* = 10 and *N* = 200, respectively. The results presented in Figure 6, Figure 7 and Figure 8 indicate that the precision values approximated the CLT values for groups with *M* = 30 and above, independent of whether the group of means (of *N* elements) had sizes of 10, 100, 200, or 500.

The asymptotic variation in the experimental values $t=\left(\bar{x}-\mu \right)/\left(s\sqrt{M}\right)$ for a population of mean *µ* and variance *σ^2^* is the Student distribution. This distribution is represented by the random variable *t*_*M*−1_. The expressions of the variables *z*($\left(\bar{x}-\mu \right)/\left(\sigma \sqrt{M}\right)=N\left(0,1\right)$) and *t* is identical when replacing the population standard deviation (*σ*) with the sample standard deviation (s). The *M* − 1 index is called the degree of freedom, and *M* is the group of means. The *M* − 1 index indicates that the shape of the distribution varies with sample size. As the degree of freedom increases, the *t_M_*_−1_ values converge quickly and then very slowly to the standard normal distribution. In the limit, with an infinite number of degrees of freedom, the *t* distribution reduces to the normal distribution. The normal variable and Student’s distribution are similar, but Student’s is more spread out for the same confidence level. Figure 9 shows the variation of the Student variable relative to normal or normalized (student variable value divided by the normal value for the same confidence level). Note that from 30 onwards, the approximation to the standard normal already takes place.

Figure 10 shows experimental results of the variation of the average of the means of several *M* values (40, 50, 60, and 70) for *N* = 10, *N* = 100, and *N* = 500 under the normalized condition. The normalized values are the uncertainties obtained experimentally divided by the standard deviation of CLT correspondent (6). The experimental uncertainties are approximately 0.001 °C, indicating that this value is the standard deviation of the measurement system.

The results shown in Figure 10 indicate that the experimental precision had a numerical value that was much greater than the variation in the mean (bias effect). The three regressions show an increasing precision as *M* increases. The uncertainty variation for *N* = 10 was smaller than *N* = 100 and *N* = 500. The bias effects (order of magnitude 1.0 × 10^− 6^°C) due to the resolution of ADC and the thermal drift of this system can become significant for *N* = 100 and *N* = 500.

The asymptotic variation in the experimental values is very close to that of the values calculated by the CLT for different values of *N*. The variation in the average, i.e., the bias effect, is much less than the precision. The instrumentation used can be improved, but it satisfies the purpose of this article.

The following analysis was based on a run of 11 consecutive days: A large number of sets of 90 elements were formed, where each element of each set was the average of *N* measurements. For each set, the variation in *M* was used to determine the mean and standard deviation. Figure 11 shows approximately 100 curves; each curve has approximately 90 data points, and each data point is a mean of 10 measurements. The values of the means are shown starting from the first data point (Figure 12).

The experimental resolution obtained from the 16-bit ADC measurements was 6.5 × 10^−4^ °C. The full range temperature (corresponding to the voltage input range (±5 V)) was 42.5984 °C. The increase in the number of bits owing to oversampling is given by (9), with the increase in bit number *w* obtained with *N* = *fos*/*fs*. The estimated standard deviation (EsSD) is given by (6). *M* is the number of elements in the set containing the mean of *N* elements. The experimental value of a given mean is equal to the set average of *M*, with *M* beginning at 40. The same applies to the standard deviation (ExSD). Variations in the averages of these two groups provided bias and experimental precision. For the bias value effect, the reference value was 20.000000 (set points).

Each set of *N* has the first, second, and third deviations, represented by *j* in Table 2 (variables *n* − *j* are obtained from sets of variables *n* − (*j* − 1)). All estimated values agreed with the experimental values, considering the resolution. Table 3 shows small variations in the central average relative to the average from the set point. Table 4 shows a small variation in the central average, considering the resolution in relation to the precision.

These results seem to indicate the validity of the proposed idea. For large values of *N*, the figures and tables seem to show some values that differ from the expected behavior but are within the experimental uncertainties. The variation of the mean for a large *N*, which relates to accuracy (the assumed true value), would correspond to the bias effect that seems to indicate it is due to ADC alone. The ADCis not yet embedded in the controlled system. From the manufacturer’s manual, the Typical Accuracy (% of Full Scale) at 15 to 35 °C to an input range ±5 V is ±0.005%. The measured temperature of 20 °C corresponds to less than 1 V. These results also seem to indicate that an ADC with greater effective resolution than the C-Field Point (AI 110), integrated into the controlled system, would give an accuracy much higher than 1 × 10^−6^ °C.

The results of the standard deviation as a function of the sample size *N* are shown in Figure 13. Figure 14 depicts the correlation between the Estimated Standard Deviation and the Experimental Standard Deviation with a correlation coefficient of 0.98 (intercept of 1.82 × 10^−6^ and slope 1.17).

An evaluation parameter for accuracy is the ratio of the standard deviation to the variation in the mean, and Figure 15 shows these good results.

## 4. Conclusions

A system that increases the accuracy and precision of the temperature meter was introduced. It consists of an embedded instrumentation and control system in which sensors and transduction electronics are immersed in a sealed environment filled with insulation oil to prevent oxidation and bias effects and to maintain a uniform temperature. Electronic temperature transduction circuits compensate for drifts and are reinforced by this configuration. It is practically characterized by the elimination of systematic effects (by the reduction of thermal, environmental, mechanical, and electrical effects) that contribute to an approximately zero-bias effect of the measurements. The applicability conditions of CLT are highlighted, and a procedure to systematically calculate the averages of the means is presented. The accuracy and precision were approximately the same for different groups of independent samples (10, 20, 50, 10, 500, 1000, or 10,000 measurements), i.e., approximately 0.001 °C, with an average of 20.000 °C. All sample groups selected at random proved the results of applying the CLT experimentally within an uncertainty determined by the theory of temporal oversampling. The increase in the number of significant digits of the measurement is proven by the increase in ADC resolution using the oversampling technique of a white noise system as a model. After hundreds of thousands of measurements, the validity of the proposed method seems to be proven, showing the possibility of applying CLT to increase accuracy and precision. Thus, an instrument with high accuracy and precision can be obtained from a system with low accuracy and precision. The idea developed can be applied to other systems with high accuracy and how Measurements of Dipole Components Content (MDCC) of a liquid sample [26] and pressure measurement for pipe flow. Despite interfering with the state of the object being measured, they are very well applied. The MDCC replaces inline the method of measuring the humidity of liquid by Karl Fischer. An MDCC performs this function inline, measuring the dipole polarization of the sample in a temperature-controlled environment with high accuracy and uniformity at static or flow conditions. In the pressure measurement, sensors, transduction electronics, and the reference pressure are in the temperature-controlled environment with high accuracy and uniformity. The pressure tapping to be measured from a flow pipe is transmitted through a duct with high horizontality.

## Figures and Tables

**Figure 1 sensors-23-03210-f001:**
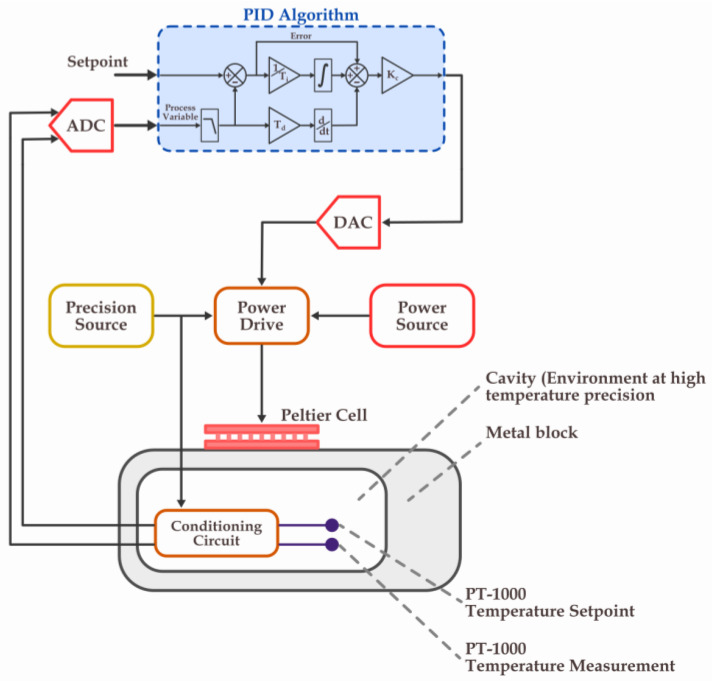
Diagram of the system used to achieve temperatures with high accuracy and precision.

**Figure 2 sensors-23-03210-f002:**
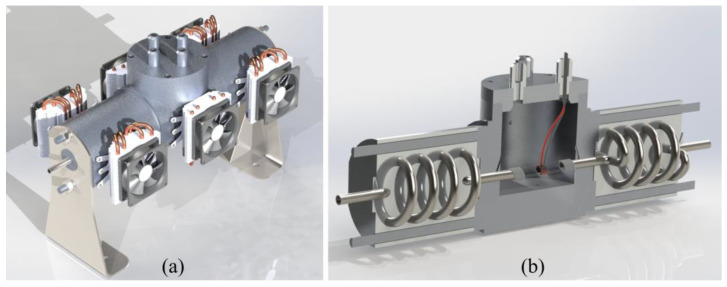
Parts of the metal block: (**a**) external view with Peltier cells and (**b**) internal view, in which the helical steel tube cast in aluminum and the environment of the sensors and electronic parts can be seen.

**Figure 3 sensors-23-03210-f003:**
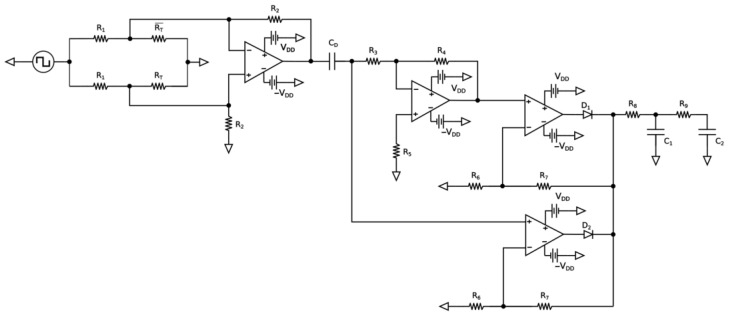
Schematic diagram of the electronic temperature measurement circuit.

**Figure 4 sensors-23-03210-f004:**
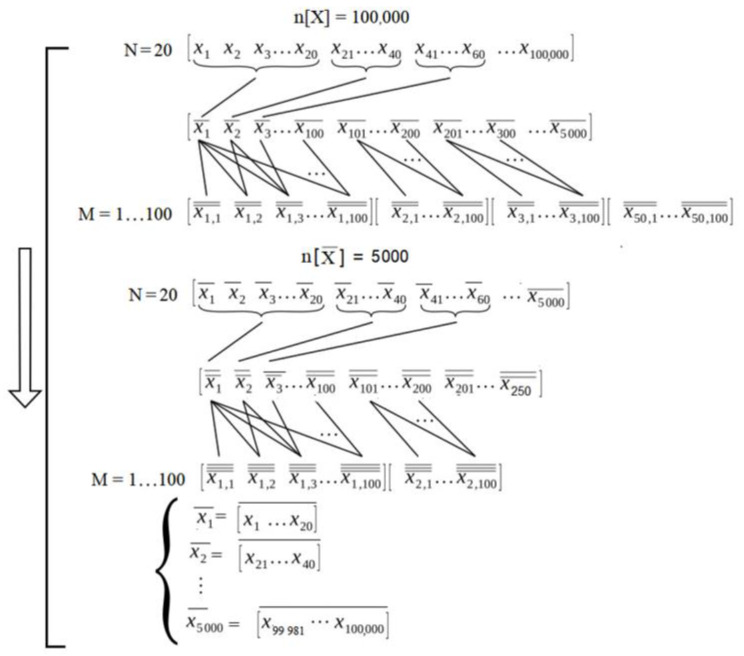
Diagram of the experimental procedure used to obtain the precision and accuracy of applying the CLT.

**Figure 5 sensors-23-03210-f005:**
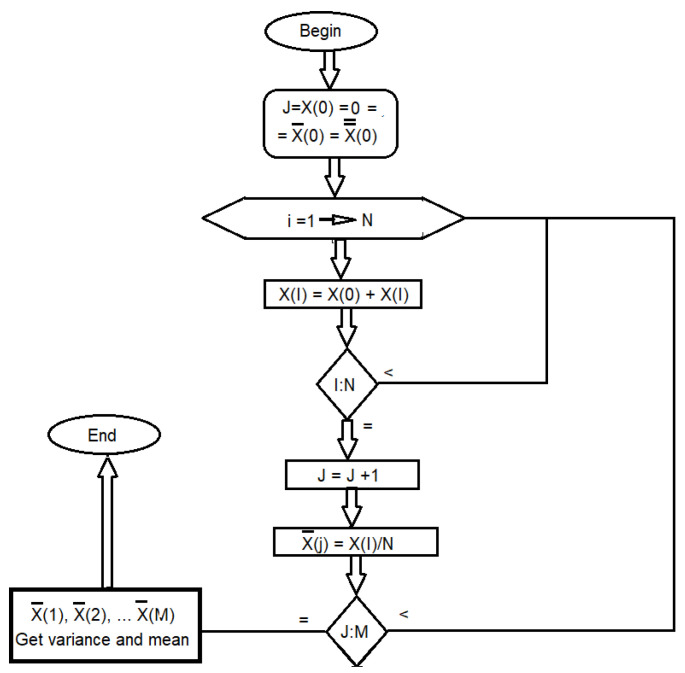
Flow chart of the acquisition of the mean and variance.

**Figure 6 sensors-23-03210-f006:**
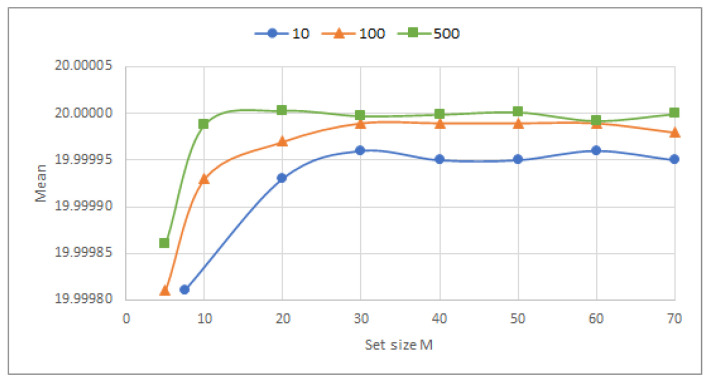
Response of several means (*N* = 10, *N* = 100 e *N* = 500) with a variation of *M* (5, 10, 20, 30, 40, 50, 60 e 70).

**Figure 7 sensors-23-03210-f007:**
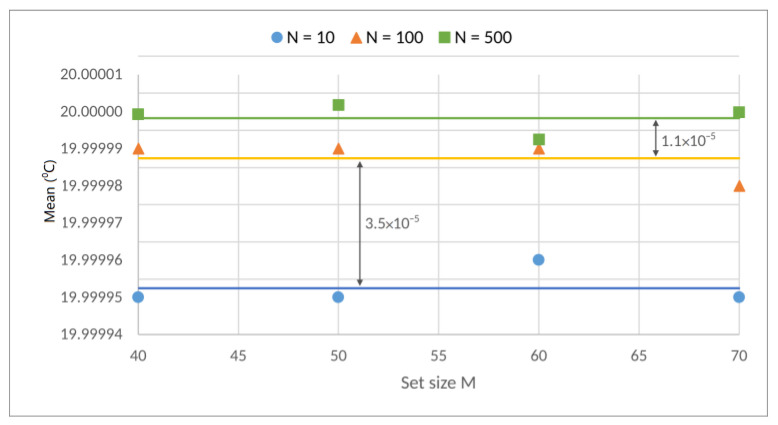
Means and their respective differences for M starting from 40.

**Figure 8 sensors-23-03210-f008:**
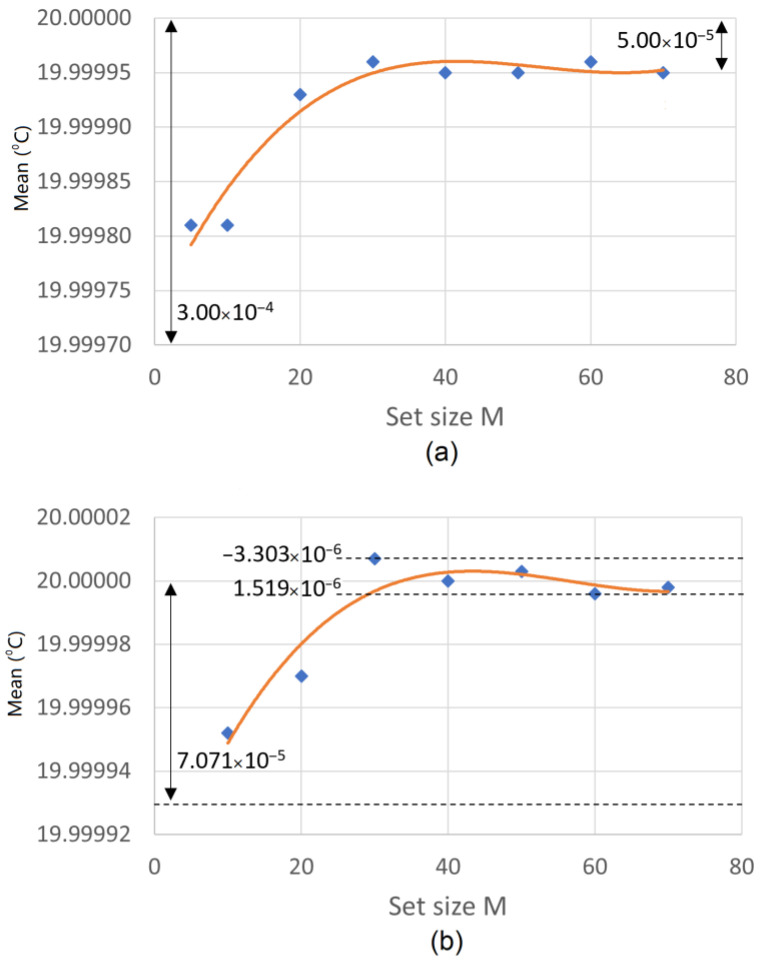
Variation in the mean and variance for (**a**) *N* = 10 and (**b**) *N* = 200.

**Figure 9 sensors-23-03210-f009:**
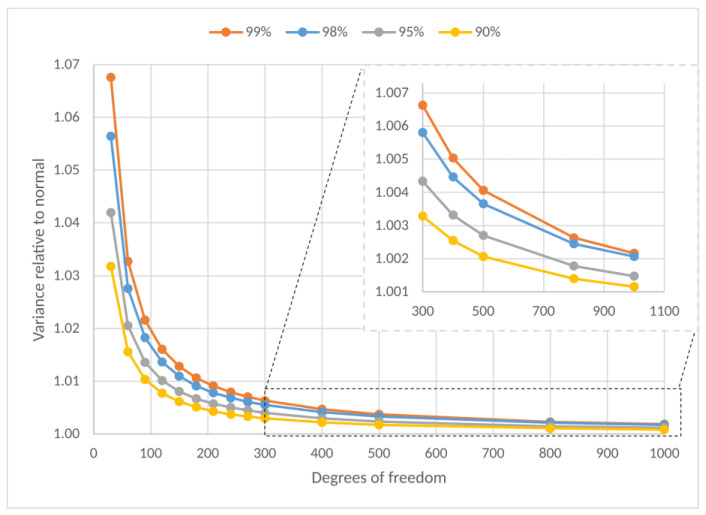
Variation of Student parameters with degrees of freedom (*M*− 1) under various probabilities starting at *M* = 30 and for groups starting at *M* = 300 (detail).

**Figure 10 sensors-23-03210-f010:**
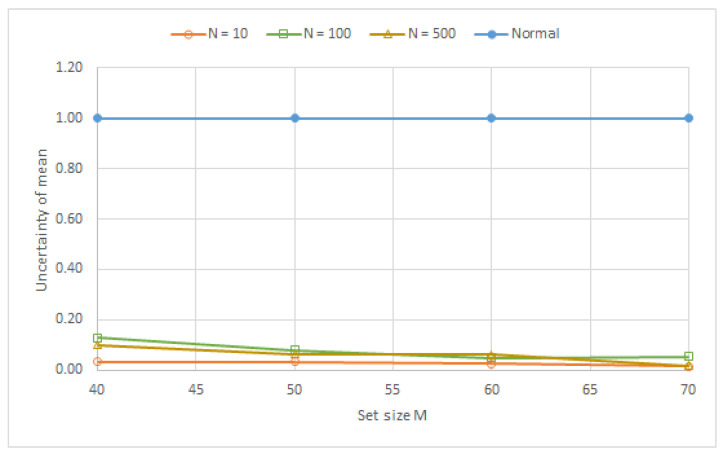
Deviation from the average relative to normal of the means.

**Figure 11 sensors-23-03210-f011:**
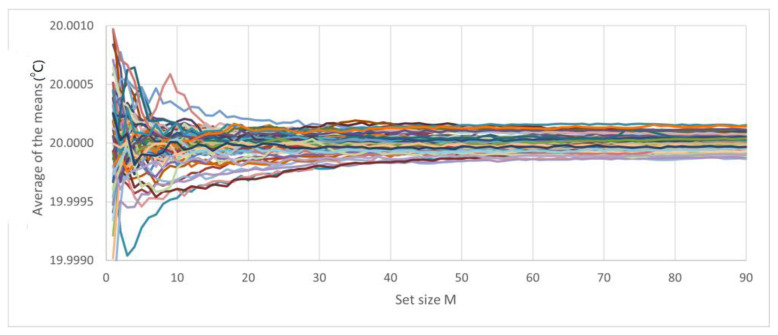
Measurements of the classes of sets of 100 elements, where each element is the average of ten measurements.

**Figure 12 sensors-23-03210-f012:**
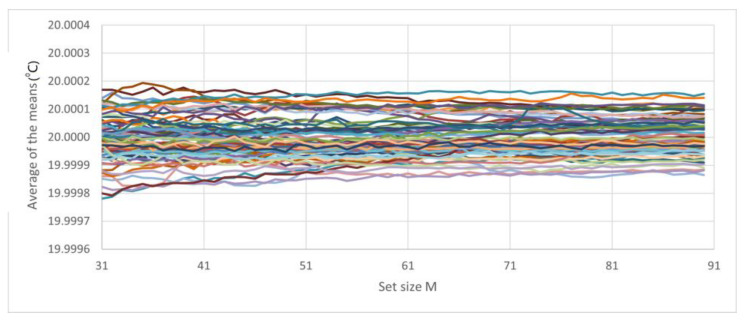
Measurements of the classes of sets of 100 elements, where each element is the average of 10 measurements, starting from *M* = 31.

**Figure 13 sensors-23-03210-f013:**
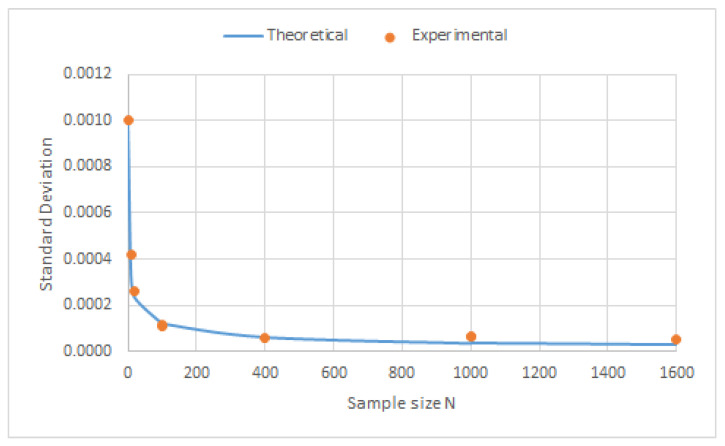
Standard deviation values as a function of the sizes of the sample mean.

**Figure 14 sensors-23-03210-f014:**
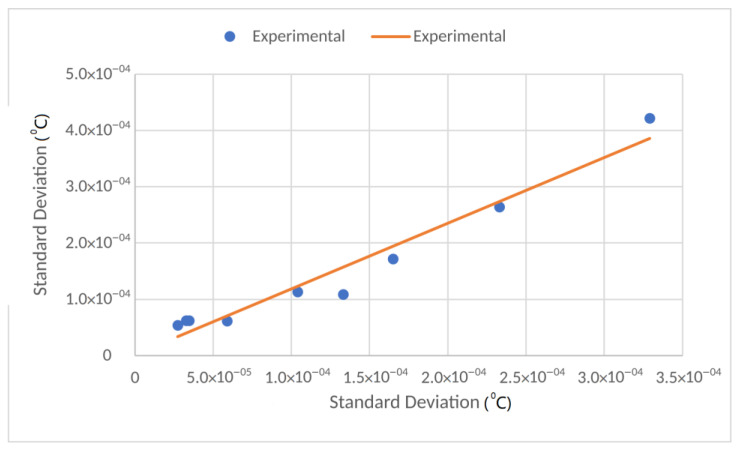
Correlation between experimental and estimated standard deviations.

**Figure 15 sensors-23-03210-f015:**
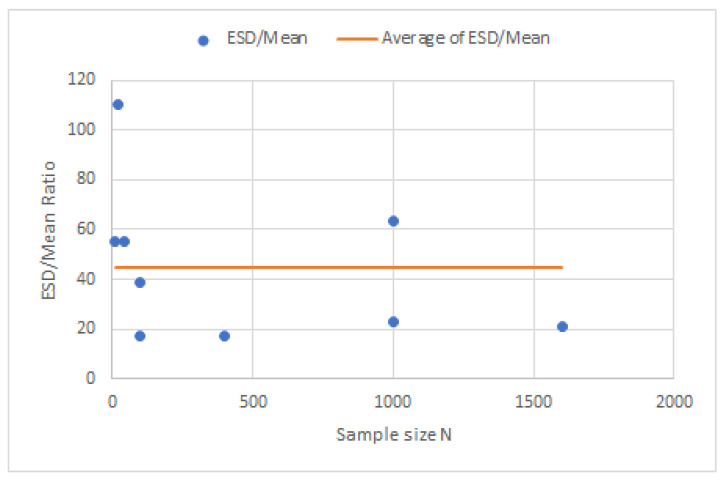
Ratio of standard deviation to many variation values of the means.

**Table 1 sensors-23-03210-t001:** Asymptotic Behavior with Increasing *M*.

M	N = 10		N = 100		N = 500	
	Mean (°C)	Δ (°C)	Mean (°C)	Δ (°C)	Mean (°C)	Δ (°C)
10	19.999813	1.87 × 10^−04^	19.999930	7.00 × 10^−05^	19.999988	1.17 × 10^−05^
20	19.999930	7.00 × 10^−05^	19.999960	4.00 × 10^−05^	20.000003	−3.30 × 10^−06^
30	19.999960	4.00 × 10^−05^	19.999987	1.30 × 10^−05^	19.999998	2.40 × 10^−06^
40	19.999950	5.05 × 10^−05^	19.999988	1.20 × 10^−05^	19.999999	8.00 × 10^−07^
50	19.999948	5.18 × 10^−05^	19.999988	1.20 × 10^−05^	20.000002	−1.70 × 10^−06^
60	19.999959	4.07 × 10^−05^	19.999988	1.20 × 10^−05^	19.999992	7.60 × 10^−06^
70	19.999952	4.77 × 10^−05^	19.999988	1.30 × 10^−05^	20.000000	2.00 × 10^−07^

**Table 2 sensors-23-03210-t002:** Mean and Variance of the average temperatures.

M_J	Resolution	EsSD ^1^ (°C)	ExSD ^2^ (°C)
10_1	2.06 × 10^−04^	3.29 × 10^−04^	4.22 × 10^−04^
10_2	6.50 × 10^−05^	1.33 × 10^−04^	1.09 × 10^−04^
10_3	2.06 × 10^−05^	3.46 × 10^−05^	6,23 × 10^−05^
20_1	1.45 × 10^−04^	2.33 × 10^−04^	2.64 × 10^−04^
20_2	3.25 × 10^−05^	5.90 × 10^−05^	6.15 × 10^−05^
40_1	1.03 × 10^−04^	1.65 × 10^−04^	1.72 × 10^−04^
40_2	1.63 × 10^−05^	2.72 × 10^−05^	5.42 × 10^−05^
100_1	6.50 × 10^−05^	1.04 × 10^−04^	1.13 × 10^−04^
1000_1	2.06 × 10^−05^	3.27 × 10^−05^	6.22 × 10^−05^

^1^ Estimated Standard Deviation. ^2^ Experimental Standard Deviation

**Table 3 sensors-23-03210-t003:** Mean and Variance of the Mean Relative to 20.000000 °C.

N_J	Resolution	Variation of the Mean (°C)	Mean Value (°C)
10_1	2.06 × 10^−04^	7.55 × 10^−06^	19.999992
10_2	6.50 × 10^−05^	1.01 × 10^−05^	19.999990
10_3	2.06 × 10^−05^	9.80 × 10^−07^	19.999999
20_1	1.45 × 10^−04^	2.00 × 10^−06^	19.999998
20_2	3.25 × 10^−05^	−3.65 × 10^−06^	20.000004
40_1	1.03 × 10^−04^	3.12 × 10^−06^	19.999997
40_2	1.63 × 10^−05^	2.65 × 10^−06^	19.999997
100_1	6.50 × 10^−05^	2.89 × 10^−06^	19.999997
1000_1	2.06 × 10^−05^	2.72 × 10^−06^	19.999997

**Table 4 sensors-23-03210-t004:** The ratio of Standard Deviation to Mean of temperatures.

Mean Variation (°C)	ExSD (°C)	Ratio of ExSD to Mean Variation
7.55 × 10^−06^	4.22 × 10^−04^	55.89
1.01 × 10^−05^	1.09 × 10^−04^	10.77
9.80 × 10^−07^	6.23 × 10^−05^	63.54
2.00 × 10^−06^	2.64 × 10^−04^	132.00
−3.65 × 10^−06^	6.15 × 10^−05^	−16.86
3.12 × 10^−06^	1.72 × 10^−04^	55.13
2.65 × 10^−06^	5.42 × 10^−05^	20.85
2.89 × 10^−06^	1.13 × 10^−04^	39.10
2.72 × 10^−06^	6.22 × 10^−05^	22.88

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
