# Peer review of "Accuracy and Precision Improvement of Temperature Measurement Using Statistical Analysis/Central Limit Theorem"

_sensors, 2023, doi:10.3390/s23063210_

Round 1

Reviewer 1 Report

The manuscript describes a method to increase the accuracy measuring the temperature of a liquid. The paper is well written and explains clearly the method used. But for me there are two critical problems:

1)    In the section two method and experimental apparatus are described, but one fundamental information (for me) is missing: the accuracy of the platinum sensor and in general the accuracy of the electronic instrumentations used. Only on page 12 and on lines 415-416 some of this information are given. In my opinion, these data should be given when the apparatus is described in section 2.

2)    The manuscript described well the different source of errors of the measurements, and the apparatus has been built to minimize as soon as possible bias, systematic and random errors. However, it is not clear if the random accuracy is below or not the sensor accuracy. Probably I’m a “old style” scientist, but for me the error of a measurement cannot be below the instrumental accuracy. People can reduce random errors, but, finally, the instrumental accuracy affects measurements, independently of the number of measurements and the methods to group them.

Consequently, I suggest the publication after two major revisions:

1)    instrumental accuracy of the electronic apparatus and the platinum sensors must be given in the description of the instrumentation.

2)    The error analysis is missing of the instrumental accuracy estimations. Please, in the discussion, the authors should give this fundamental value and compare it with the random error that they found, and explain clearly that the final error of the measurement depend only on the instrumental accuracy, because the measurements have been taken to reduce as soon as possible the other sources of errors. 

Author Response

Dear reviewer,

Thank you very much for the proposed questions They undoubtedly contributed to improving the paper.

We think that your questions were answered according attachment text.

Please see the attached for details.

Best Regards,

Prof. Belo

Reviewer 2 Report

Based on my understanding, I think generally this paper is mainly a statistical analysis paper where by increasing the number or samples that is without bias will eventually provide better data accuracy and precision. I was a bit misled by the title of the paper initially as I was expecting to see some system design for temperature measurement.

If I may, I would suggest to delete the word "System" and change the title of the paper to "Accuracy and precision improvement of temperature measurement using statistical analysis/ central limit theorem".

1. Too many definition of terminology in introduction but failed to highlight how CLT can be better in improving the accuracy and precision on temperature measurement. 

2. Should provide details information about the targeted temperature measurement, range, purposes or processes involved.

3. Abstract mentioned in line 20, "special system". Please justify. 

4. Page 2, line 77. "This is wrong." this should be rewritten properly. 

5. Before Section 2 on the materials and methods and the equation for the CTL, suggest to provide an overall picture or diagram showing where these equations will be implemented? It is after data acquisition that is not within the temperature measurement system? Because Figure 1 in page 5, only have digital processor that based on PID controller. 

6. Page 8 line 315 to 328, suggest to provide flow chart and then explain the flow in paragraph. 

7. I am lost at page 10 line 348 with the student parameters. What do you mean? or it is just the same meaning for the M=30 group? The student parameters continue in Figure 8 and 9. Please explain.

8. page 11, line 371, justify why the precision and bias is caused by the DAC?

9. all result and graph figures should be standardised with appropriate legend and proper axis labels.

10. Please check carefully for typo and errors on the manuscript.

11. References can be further updated. Four of the most recent references are from the same authors group, maybe can include some other recent and related papers?

Author Response

Dear Reviewer,

Thank you very much for the proposed questions. They undoubtedly contributed to improving the paper.

We think that your questions were answered according attachment text.

Please see the attached document for details.

Best Regards,

Prof. Belo

Reviewer 3 Report

Dear authors, the paper presents a well-conducted study with a complex amount of data; therefore, I feel you could improve the paper by clarifying several aspects summarised below.

Major comments

1) The introduction needs major English proofing. 

2) Some paragraphs from the Introduction could be better suited for the Discussion section. 

eg.: "Experimental results from hundreds of thousands of measurement data points seem to validate the proposed method in the presence of the most varied effects due to systematic phenomena applied to the measurement of the temperature."

3) Some paragraphs from the Introduction could be better suited for the Conclusion section. 

eg.: "The idea developed can be applied to other systems with high accuracy how Measurements of Dipole Components Content (MDCC) of liquid sample [13] and pressure measurement for pipe flow. In spite to interfere in the state of the object being measured, are very well applied."

4) There is no Discussion section where you may be worth mentioning the results obtained up until now and how your results can improve the measurement system.

5) The References are inconsistently introduced and lack the journal's recommendations; from 18 references, 4 are self-citations.

Minor comments

- Lines 65 and 67 are the same: "h) Approximations and assumptions incorporated in the measurement method and procedure; i) Approximations and assumptions incorporated in the measurement method and procedure.”

-inconsistent add notations throughout the text eg: "Measurements of Dipole Components Content (MDCC)" and then "IID (independent and identically distributed)"

Author Response

Dear Reviewer,

Thank you very much for the proposed questions They undoubtedly contributed to improving the paper.

Please see the attached documents for details.

Best regards,

Prof. Belo

Round 2

Reviewer 1 Report

The manuscript describes a method to increase the accuracy measuring the temperature of a liquid. This the second version of the paper. In the first version, I asked two improvements: 1) specify better the instrumental accuracy; 2) analyze the experimental errors also considering the instrumental accuracy. In this second version both these two requests have been satisfied: authors explain exhaustively the sources of instrumental errors and in the analysis of the errors also consider this source.

There are only two typographic corrections to do before publications:

Lines 330-333: pay attention to the text size.

Line 334: 0.001 0C : should be 0.001 °C.

Author Response

Dear reviewer,

Thank you very much.

I think that your questions were answered. You can see this in the document.

Best Regards,

Prof. Belo

Reviewer 2 Report

The paper has been improved significantly. However please check for typos thoroughly. 

1. example of typo: page 7, line 365 typo eeliminatingtherman, line 267 typo bthe ridge

2. Table 2, is it possible to indicate the M_? for the first column?

Author Response

(The authors gave the same response as above.)

Reviewer 3 Report

After the revision, the author managed to improve the paper considerably. 

There are minor issues I consider could be addressed:

CTL- full name in the abstract but not in Introduction

261- space between citation and 2020

265- spelling mistake

267- spelling mistake

280- rephasing could improve the intelligibility of the text

330-333- the paragraph has a different font

427- "The M-1" has a supplementary space

460- spelling mistake

461- dot before the beginning of a paragraph

511- punctuation regarding Figures

- references are inconsistently introduced

Author Response

(The authors gave the same response as above.)
